# High-Grade Thalamic Glioma: Case Report with Literature Review

**DOI:** 10.3390/medicina60101667

**Published:** 2024-10-11

**Authors:** Corneliu Toader, Mugurel Petrinel Radoi, Adrian Dumitru, Luca-Andrei Glavan, Razvan-Adrian Covache-Busuioc, Andrei Adrian Popa, Horia-Petre Costin, Antonio-Daniel Corlatescu, Alexandru Vladimir Ciurea

**Affiliations:** 1Department of Neurosurgery, “Carol Davila” University of Medicine and Pharmacy, 020021 Bucharest, Romania; corneliu.toader@umfcd.ro (C.T.); luca-andrei.glavan0720@stud.umfcd.ro (L.-A.G.); razvan-adrian.covache-busuioc0720@stud.umfcd.ro (R.-A.C.-B.); andreiadrianpopa@stud.umfcd.ro (A.A.P.); horia-petre.costin0720@stud.umfcd.ro (H.-P.C.); antonio.corlatescu0920@stud.umfcd.ro (A.-D.C.); prof.avciurea@gmail.com (A.V.C.); 2Department of Neurosurgery, National Institute of Neurology and Neurovascular Diseases, 020021 Bucharest, Romania; 3Department of Pathology, “Carol Davila” University of Medicine and Pharmacy, 020021 Bucharest, Romania; 4Department of Pathology, Emergency University Hospital Bucharest, 050098 Bucharest, Romania; 5Department of Neurosurgery, Sanador Clinical Hospital, 011038 Bucharest, Romania

**Keywords:** prognostic factors, microsurgical excision, high-grade gliomas, neurosurgical interventions, thalamic gliomas

## Abstract

This case report delves into the case of a 56-year-old female patient presenting with progressive cephalalgia syndrome, nausea, vomiting, and gait disorders, diagnosed with a high-grade thalamic glioma. Glioma is the most common form of central nervous system (CNS) neoplasm that originates from glial cells. Gliomas are diffusely infiltrative tumors that affect the surrounding brain tissue. Glioblastoma is the most malignant type, while pilocytic astrocytomas are the least malignant brain tumors. In the past, these diffuse gliomas were classified into different subtypes and grades based on histopathologies such as a diffuse astrocytoma, oligodendrogliomas, or mixed gliomas/oligoastrocytomas. Currently, gliomas are classified based on molecular and genetic markers. After the gross total resection, a postoperative brain CT scan was conducted, which confirmed the quasi-complete resection of the tumor. The successful gross total resection of the tumor in this case, coupled with significant neurological improvement postoperatively, illustrates the potential benefits of aggressive surgical management for thalamic gliomas. This report advocates for further research to assess the efficacy of such interventions in malignant cases and to establish standardized treatment protocols, considering the heterogeneity in prognostic outcomes and the advancements in molecular diagnostics that offer deeper insights into glioma oncogenesis and progression.

## 1. Introduction

Thalamic neoplasms account for about 1% of all brain tumors [1]. Their deep location near critical brain areas has made surgery challenging.

A particular study reported a male-to-female ratio of approximately 1.3:1 for adult thalamic gliomas, indicating a moderate prevalence in males [2].

However, improvements in neurosurgical techniques and tools have significantly reduced the risks and complications of surgery for thalamic gliomas [3]. Yet, survival outcomes after surgery vary widely, highlighting the need to identify key prognostic factors for patients, especially those with high-grade thalamic gliomas [4]. Thalamic gliomas predominantly affect pediatric and adolescent populations, though they are not confined to these age groups. Astrocytomas constitute around 88% of primary thalamic neoplasms [5]. Pediatric cases of thalamic glioma generally exhibit a more favorable histology in contrast to the predominantly malignant histological profile observed in adult cases. The therapeutic approach to thalamic glioma has evolved significantly over the past two to three decades, transitioning from initial radiation therapy without a preceding tissue diagnosis to more refined techniques such as open biopsy or partial resection, stereotactic biopsy, and ultimately, microsurgical excision [6]. This evolution in surgical strategy, particularly the shift towards more aggressive resections, has been reported to enhance survival rates, especially in patients with low-grade gliomas. However, the efficacy of aggressive surgical interventions in cases of malignant thalamic gliomas remains to be fully assessed [7,8]. Relative to non-thalamic gliomas, thalamic gliomas are markedly less common. Traditionally, due to the challenges posed by the thalamic location, radical resection of these tumors has been deemed problematic, with diagnostic biopsies often preferred. Consequently, a standardized treatment protocol for thalamic gliomas remains elusive, and the subject has not been extensively researched [9]. The complexity of treating thalamic gliomas arises from several factors: primarily, the thalamus is situated in a deeply embedded region within the supratentorial brain, characterized by intricate anatomical structures, complicating surgical access and intervention. Furthermore, thalamic gliomas typically exhibit heightened malignancy and are associated with severe postoperative complications [10]. Over the past two decades, significant advancements in the diagnosis and treatment of gliomas have been achieved. It has been established that molecular diagnostics for gliomas offer more substantial insights compared to sole reliance on histological evaluations, enhancing diagnostic accuracy and prognostic predictions. The IDH1 gene, responsible for encoding isocitrate dehydrogenase 1, plays a crucial role in the oxidative carboxylation of isocitrate, leading to the production of α-ketoglutarate. Mutations in IDH1 result in the elevated synthesis of R-2-hydroxyglutarate (R-2-HG), a metabolite implicated in the oncogenesis and progression of gliomas. R-2-HG is also gaining recognition as a potent predictive biomarker for gliomas, signifying a unique subclass of these tumors characterized by specific oncogenic mechanisms [11].

Regarding the IDH status in thalamic tumors, current studies indicate that mutations in the IDH1 and IDH2 genes are relatively uncommon in thalamic gliomas compared to their prevalence in gliomas located in other regions, such as the frontal lobe. Thalamic gliomas with IDH mutations are rare and may demonstrate distinct clinical and biological characteristics when compared to their IDH wild-type counterparts [12].

Additionally, other genetic mutations are known to be present in thalamic tumors. Notably, the H3F3A K27M mutation is frequently identified in thalamic gliomas, particularly in pediatric patients [13].

This mutation characterizes a unique subset of high-grade gliomas associated with a more aggressive clinical course. Other genetic alterations, including mutations in TP53, ATRX, and EGFR, are also implicated in the pathogenesis of thalamic tumors, contributing to their distinctive molecular profile compared to gliomas in other areas of the brain [14].

These findings highlight the genetic diversity within thalamic gliomas and underscore the need for further molecular characterization to better understand their behavior and potential therapeutic targets.

## 2. Case Presentation

56-year-old female patient presented in our clinic with persistent cephalalgia, nausea, vomiting, and gait disorders (gait apraxia) occurring for approximately 2 months with progressive intensification of symptoms. Neurological examination revealed a syndrome of intracranial hypertension, right central type facial paresis, and right hemiparesis predominantly brachial. On objective examination, the patient showed clinically normal lungs and cardiovascular assessment showed an arterial hypertension grade II and hypertensive retinal angiopathy stage II. A cranial computed tomography (CT) scan was conducted, which disclosed the presence of a neoplasm located in the right thalamus, measuring 5.5 by 6 cm. This lesion exhibited a hypodensity relative to the surrounding cerebral tissue in the pre-contrast phase and demonstrated moderate inhomogeneous enhancement following the administration of an iodinated contrast agent. There was a significant mass effect on the third ventricle, accompanied by a notable midline shift of approximately 1 cm, indicative of subfalcine herniation. Apart from the described pathology, the cerebroventricular system presented a normal tomodensitometric appearance. Native brain MRI with paramagnetic substance highlighted the previously described tumor formation, hypointense in T1 and hyperintense in T2 with discrete and inhomogeneous intake after administration of gadolinium DTPA and infiltrative character suggestive of a high-grade glioma. Otherwise, the normal appearance of the brain was highlighted in the T1 and T2 weighted sequences and after administration of paramagnetic substance (Figure 1, Figure 2 and Figure 3). The cystic component of the lesion was characterized by a cyst containing fluid of a yellowish hue and oily consistency, exhibiting a density exceeding that of cerebrospinal fluid, indicative of its hematogenous origin (absorbed blood). The solid portion is highly vascularized, presenting a reddish coloration and possessing a soft texture amenable to aspiration.

Surgery was performed on the tumor, and gross total tumor resection was achieved, utilizing a parieto-occipital surgical approach on the designated side. The transcortical access was used to approach the neoplasm, subsequently followed by stepwise ablation under magnification with microsurgical tools. Efforts were made to preserve the vascular structures at the specific level, including the thalamostriate veins along with their tributaries, and the perforating branches emanating from the posterior cerebral artery that penetrate the tumor. A postoperative brain CT scan was conducted, which confirmed the quasi-complete resection of the tumor and an area of right thalamic hypodensity and right deep parietal of sequelae aspect. Otherwise, there was a normal cerebroventricular tomodensitometric aspect. There were no signs of bleeding in the tumor bed (Figure 4).

Postoperatively, the patient presented without significant neurological impairments, exhibiting only minimal sensory deficits on the left side, absent paresis, and left-sided hemihypesthesia. A treatment regimen comprising whole-brain radiotherapy alongside chemotherapy with temozolomide was initiated. During the initial treatment cycle, both modalities were employed concurrently, as radiotherapy is known to enhance the permeability of the blood–brain barrier to temozolomide, thereby facilitating its therapeutic efficacy.

Histopathological examination revealed the dense cellularity, nuclear pleomorphism, and areas of necrosis were consistent with aggressive tumor behavior, often seen in high-grade astrocytomas. These necrotic regions are frequently surrounded by pseudopalisading cells, a key histopathological feature of glioblastomas (Figure 5) [15].

Loss of ATRX expression, often coupled with IDH1 mutations, is common in gliomas and is associated with specific subtypes such as astrocytomas. ATRX loss correlates with chromosomal instability, contributing to tumor progression [16].

IDH1 mutations are frequently associated with secondary glioblastomas and confer a better prognosis compared to IDH1 wild-type gliomas. These mutations are critical for determining the molecular classification of gliomas, influencing treatment strategies [17].

The lack of vimentin staining in this oligodendroglioma-like cell population may indicate a shift towards oligodendroglial differentiation. Vimentin, typically a marker for mesenchymal cells, is commonly found in astrocytic tumors but may be absent or less prominent in tumors exhibiting oligodendroglial characteristics [18].

## 3. Discussion

The WHO 2021 Classification of Central Nervous System Tumors delineates gliomas based on their predominant occurrence in either adults (termed “adult-type”) or children (“pediatric type”), acknowledging that pediatric tumors can manifest in adults, particularly young adults, and adult tumors may infrequently be observed in children. In this updated classification, adult-type diffuse gliomas are categorized into three distinct groups: astrocytoma with IDH mutation, oligodendroglioma characterized by IDH mutation and 1p/19q co-deletion, and glioblastoma with IDH wild-type status. Pediatric-type diffuse gliomas are less common, with circumscribed gliomas and glioneuronal tumors such as pilocytic astrocytoma and ganglioglioma being more prevalent in this group [19]. Low-grade diffuse astrocytomas are distinguished by IDH mutations, and as per the WHO CNS5, their prognosis is anticipated to be more favorable compared to the classification under WHO 2016. Given the similar prognostic outlook for IDH-mutant grade 2 and 3 astrocytomas, the recent literature tends to classify these together as “diffuse low-grade astrocytomas.” Conversely, the practice of aggregating “high-grade astrocytomas” (grades 3 and 4) is now advised against due to the distinct molecular profiles and clinical trajectories of IDH-mutant grade 3 astrocytomas compared to IDH-wild-type grade 4 astrocytomas, such as glioblastoma [19,20]. Astrocytoma IDH-mutant is now recognized as a singular tumor entity, graded as CNS WHO 2, 3, or 4, with the term “anaplastic” being omitted. Accordingly, astrocytoma IDH-mutant CNS WHO grade 3 is the designated nomenclature for this classification [21,22]. Glioblastoma IDH-wild-type CNS WHO grade 4 is characterized by the presence of necrosis and/or microvascular proliferation. It has been noted that IDH-wild-type astrocytomas, classified histopathologically as grades 2 or 3 due to the absence of necrosis or microvascular proliferation, exhibit clinical behaviors akin to glioblastomas [23].

Extensive research indicates that glioblastoma multiforme may contain cells derived from astrocyte-like neural stem cells located within the SVZ (Subventricular Zone), situated just beneath the ependymal layer of the brain ventricles [24,25]. Lim et al. have observed that GBMs in contact with the SVZ are more likely to present as multifocal at the time of diagnosis [26]. The literature outlines that the choice of surgical approach for tumor resection is influenced by the tumor’s epicenter and its proximity to the corticospinal tract (CST) [27,28]. The most direct route from the cortex to the tumor is selected, taking into account the tumor’s spread pattern and CST location [27]. A transcortical approach is utilized for tumors located in the antero- or posterolateral thalamus that extend in a superior lateral direction. Tumors in the posterolateral thalamus warrant a transcortical–transventricular approach. For tumors situated in the medial and posterior superior thalamus, an interhemispheric transcallosal approach is preferred [29]. Lateral thalamic lesions are addressed with a trans-sylvian–transinsular approach, while posterior inferior and medial posterior inferior thalamic lesions are approached via an occipital transtentorial method [30]. Anterior thalamic tumors are treated using a modified lateral supraorbital (MLSO) approach [31].

Thalamic astrocytomas have the potential to cause epilepsy, although this is less frequent compared to astrocytomas in regions such as the temporal or frontal lobes. Nonetheless, seizures may still occur due to the involvement of the thalamus in regulating neural activity [32].

Seizures associated with thalamic tumors may arise due to the tumor’s impact on adjacent brain structures or its involvement in thalamocortical circuits, which are crucial for maintaining cortical excitability [33].

Thalamic astrocytomas are often associated with a poorer prognosis due to their deep location, which poses challenges for complete surgical resection. Their proximity to vital structures makes them more challenging to treat surgically, and they may exhibit a different pattern of growth compared to astrocytomas in more accessible areas of the brain [5].

This unique localization also influences the types of symptoms experienced by patients and may require tailored therapeutic approaches.

The established treatment for glioblastoma multiforme (GBM) encompasses surgical intervention, concurrent temozolomide chemotherapy with radiotherapy, followed by six cycles of adjuvant temozolomide. Data from the EORTC trial EORTC-26981/CAN-NCIC-CE3 revealed that GBM patients exhibited a 6-month progression-free survival (PFS) rate of 53%, a median survival of 14.6 months, and a 2-year survival rate of 26.5% [34]. However, the optimal duration of adjuvant chemotherapy in GBM patients remains undetermined, leading to an increasing number of clinicians extending temozolomide therapy beyond the initial six cycles. Balañá et al. reported that in Spain, 80.5% of neuro-oncologists continued temozolomide therapy for more than six cycles [35]. Currently, several long-term temozolomide treatments are employed, including the standard ‘5/28’ protocol (150 to 200 mg/m^2^ on days 1 to 5 of a 28-day cycle), the ‘1 week on-1 week off’ protocol (150 mg/m^2^ on days 1 to 7 and 15 to 21 of a 28-day cycle), and the dose-dense regimen (75 to 100 mg/m^2^ on days 1 to 21 of a 28-day cycle) [36,37,38]. In cases of recurrent glioblastoma multiforme where surgery is not viable, alternative treatments such as stereotactic radiosurgery (SRS) and supplementary chemotherapy are considered. The effectiveness and safety of SRS have been variably reported, though it seems to offer potential benefits to specific patient subgroups, particularly those with limited focal recurrences [39]. Within the recurrent GBM population, Friedman et al. reported an overall survival of 9.2 months following bevacizumab treatment [40], while Desjardins et al. documented an OS of 9.3 months with a combination of bevacizumab and temozolomide [40]. Other research has consistently presented similar or reduced survival durations following various chemotherapeutic approaches for GBM recurrence.

Recent genomic studies have identified unique genetic alterations that delineate distinct subsets of glial tumors, influenced by factors such as tumor classification, patient age, and tumor origin. Notably, novel histone mutations have been identified in thalamic tumors, predominantly in pediatric and young adult populations [41,42]. Although diffuse midline gliomas harboring the histone H3-K27M mutation are characterized by aggressive clinical manifestations and unfavorable outcomes, recent investigations suggest that this mutation in thalamic gliomas of adult patients may not necessarily correlate with a poorer prognosis compared to those without the mutation [13,43,44]. This indicates a degree of molecular heterogeneity within this group, underscoring the need for further extensive studies to assess the therapeutic potential of targeting histone-modifying enzymes in adult thalamic glioblastoma cases [27,45]. While studying the literature on thalamic gliomas, we have delved into studies that presented significant contributions to the field, some of them being listed in Table 1.

Multiple studies examine the complexities involved in the surgical management of thalamic gliomas, consistently highlighting the challenges of achieving complete resection due to their difficult anatomical location. The research collectively indicates that, even with the use of advanced surgical techniques, these tumors are frequently associated with high morbidity and poor prognosis [6].

Several studies have examined the surgical outcomes of thalamic gliomas, emphasizing that limited access to these deep-seated tumors and the high risk of postoperative complications are key factors contributing to their generally poor prognosis [49].

## 4. Conclusions

In wrapping up our debate about this patient’s case, this study highlighted various significant aspects of the pathology of high-grade thalamic gliomas. Preoperative neurological deficits were important factors that guided the neurosurgeon to an adequate prognosis. The special location, considerable size, and histopathological type of glioma make this case a real challenge for a neurosurgeon. Moreover, after gross tumoral resection, the patient experienced a favorable postoperative outcome with no significant neurological impairments. The individual displayed only minimal sensory deficits on the left side without paresis and experienced left-sided hemihypesthesia. The treatment regimen was initiated, incorporating whole-brain radiotherapy combined with chemotherapy using temozolomide. This approach was particularly chosen because radiotherapy has been shown to increase the permeability of the blood–brain barrier to temozolomide, enhancing the drug’s efficacy. This case illustrates the effectiveness of integrating surgical intervention with a targeted postoperative treatment plan to manage brain-tumor-related conditions, emphasizing the importance of tailored therapeutic strategies to optimize patient outcomes.

## Figures and Tables

**Figure 1 medicina-60-01667-f001:**
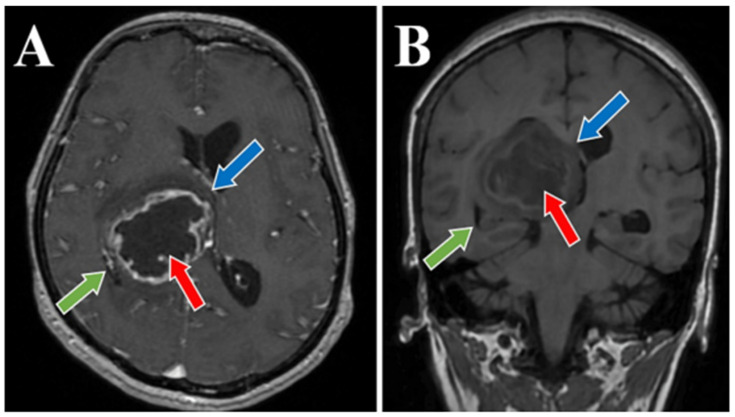
Preoperative MRI T1 and T1 Gd sequence, axial and coronal section. MRI T1 Gd-sequence axial section (**A**) and MRI T1 coronal section (**B**) both depict a thalamic glioma (red arrows), with a diameter of 4.5 cm in figure (**A**) and 4.8 cm in figure (**B**). Moreover, an important displacement of about 1 cm of the midline, with signs of subfalcine engagement (blue arrows) and infiltrative character suggestive of a high-grade glioma (green arrows) is present in both images.

**Figure 2 medicina-60-01667-f002:**
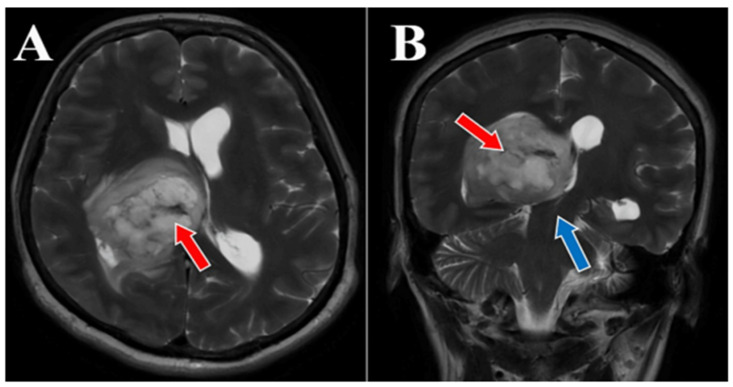
Preoperative MRI T2 sequence, axial and coronal section. MRI T2-sequence axial section (**A**) and coronal section (**B**) both depict a thalamic glioma with various morphological aspects (red arrows) with marked mass effect on the third ventricle (blue arrow). The hyperintensity on T2-weighted images with heterogeneous and subtle enhancement following the administration of gadolinium DTPA exhibits an infiltrative nature suggestive of a high-grade glioma. The glioma measured 5.5 × 6 cm, associated with a significant midline shift of around 1 cm.

**Figure 3 medicina-60-01667-f003:**
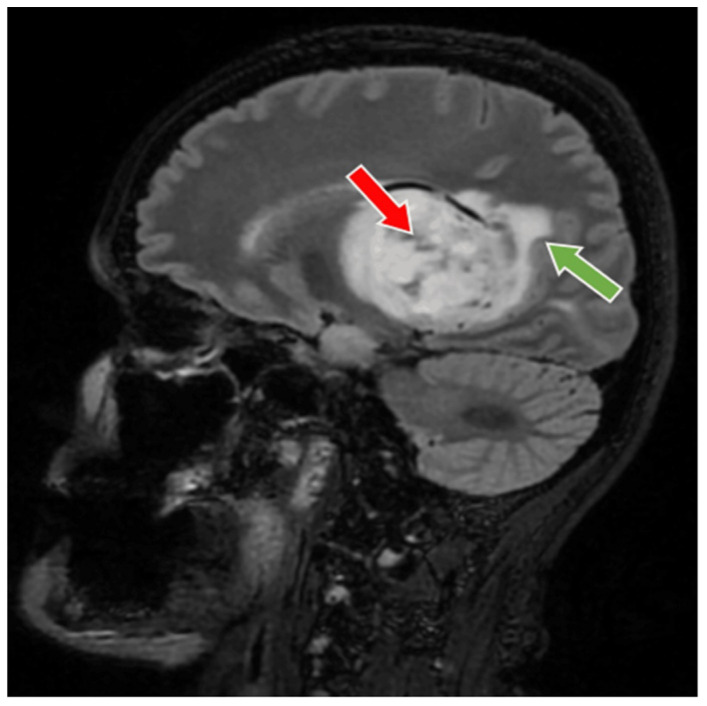
Preoperative MRI T2-FLAIR sequence, sagital section. MRI T2-FLAIR sequence sagital section depict a thalamic glioma (red arrow), with an infiltrative character suggestive of a high-grade glioma (green arrow). The glioma has a diameter of 6 cm.

**Figure 4 medicina-60-01667-f004:**
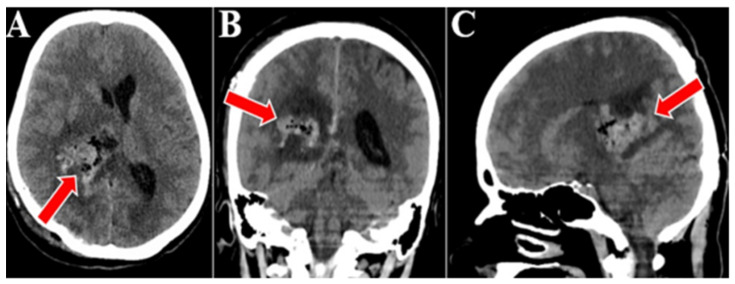
Postoperative CT scan, all three sequences, axial (**A**), coronal (**B**), and sagital (**C**). Postoperative CT scan that confirmed the gross total resection of the tumor, an area of right thalamic hypodensity and right deep parietal of sequelae aspect (red arrows).

**Figure 5 medicina-60-01667-f005:**
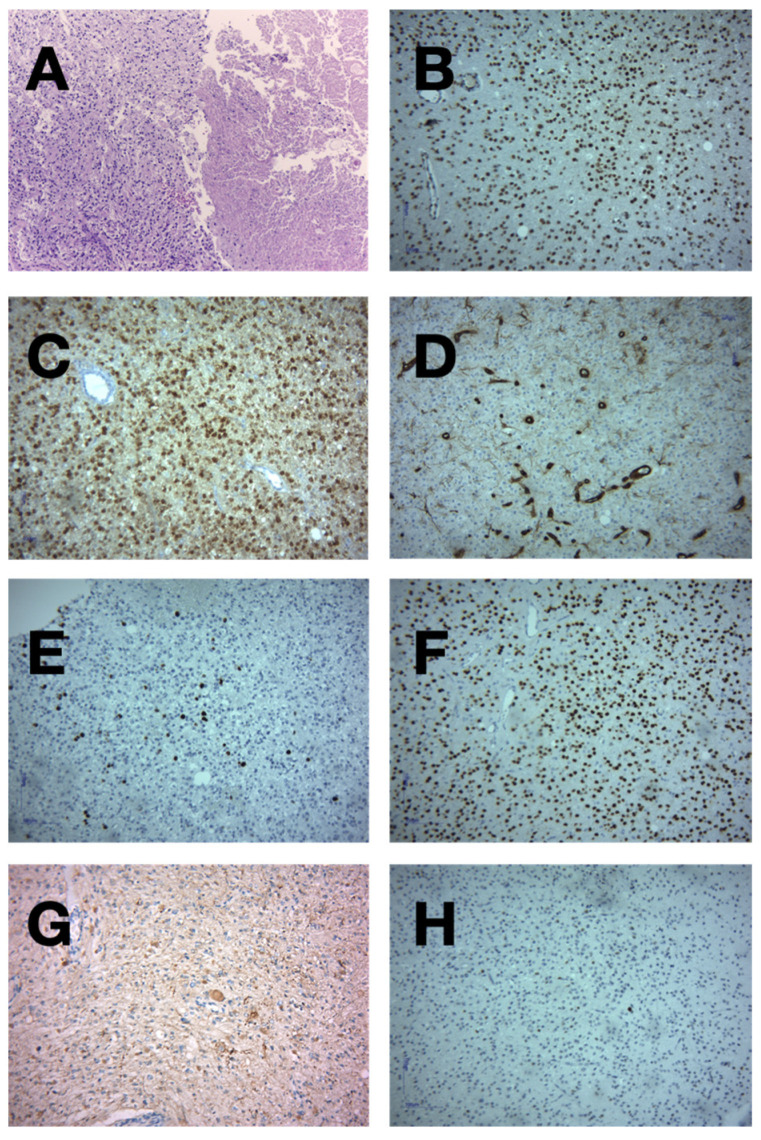
These images obtained after anatomic pathologic examination provide a comprehensive morphological and molecular overview of high-grade gliomas, supporting their classification and potential molecular pathogenesis. Section (**A**): Hematoxylin and Eosin stained section of a high-grade glioma with necrosis and cellular pleomorphism. Section (**B**): ATRX expression in high-grade glioma. Section (**C**): IDH1 mutation expression in high-grade glioma. Section (**D**): Negative Vimentin expression in oligodendroglioma-like cells within a high-grade glioma. Section (**E**): The positive nuclear staining is indicated by the brown coloration within the cell nuclei, suggesting active proliferation. The intensity of the staining is moderate, and there appears to be a moderate density of stained nuclei. Section (**F**): The image shows higher nuclear positivity compared to the left, indicating a higher proliferation rate, typical of more aggressive tumors. The homogeneous distribution of positive nuclei suggests high tumor cellularity and malignancy, which is characteristic of high-grade gliomas. Section (**G**): The brown cytoplasmic staining indicates vimentin-positive cells, which are more typical of astrocytic differentiation. The presence of vimentin in this image may suggest a mesenchymal or astrocytic component of the tumor, often associated with more invasive or aggressive behavior in gliomas. Section (**H**): This absence of staining might indicate the presence of an oligodendroglioma, which typically lacks vimentin expression, supporting differentiation toward the oligodendroglial lineage.

**Table 1 medicina-60-01667-t001:** Studies emphasizing microsurgical treatment of thalamic gliomas. No = number of patients; F/M = female/male patients; GTR = gross total resection; STR = subtotal resection; PR = partial resection; NA = Not available.

First Author and Year	Age (Median)	Sex, No of Patients		Tumor Volume (Median)	Extent of Resection			Neurological Complications			Surgical Complications	
		M	F		GTR	STR	PR	Motor Deficits	Sensory Deficits	Visual Deficit	Hemorrhage	Hydrocephalus
**Sai Kiran NA et al., 2013** [27]	29	15	7	NA	9	13	16	16	NA	NA	0	9
**Lim J et al., 2021** [29]	42	19 patients		26 cm^3^	11	7	6	6	3	3	5	3
**Esquenazi et al., 2018** [46]	53	31	26	13 cm^3^	0	57	35	35	14	14	5	27
**Niu X et al., 2020** [47]	41	56	46	4 cm^3^	46	50	NA	NA	NA	NA	4	11
**Nishio S et al., 1997** [48]	24	11	9	NA	1	0	5	2	3	3	NA	NA

## Data Availability

The data presented in this study are available on request from the corresponding author.

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
