# Peer review of "High-Grade Thalamic Glioma: Case Report with Literature Review"

_medicina, 2024, doi:10.3390/medicina60101667_

Round 1
Reviewer 1 Report
Comments and Suggestions for Authors
In the article by Wong TT et al., 2016 (10.1007/s00381-016-3148-5), a comprehensive description of thalamic tumours in children is provided. It is estimated that the prevalence of this particular type of tumour is approximately 4% of all pediatric brain tumours. In the manuscript, a clinical case of a 56-year-old female patient is presented for consideration. What is the incidence of thalamic tumours in women?
The introduction concludes with a description of the IDH mutation and a concise account of its mechanism. It would be beneficial to expand this section and, if we are discussing mutations, to include information on the IDH status of thalamic tumours. Are additional mutations also present, or do they differ only in location?
On page 5, the abbreviation "SVZ" is mentioned, but it is never defined in the manuscript.
The article is an interesting report of a successful surgical resection of an astrocytoma in the thalamus, accompanied by a concise review of the relevant literature. The surgical techniques employed in this case are described in detail. However, the remainder of the review is devoted to astrocytomas in general, without focusing on the specific characteristics of those that originate in the thalamus. It would be beneficial to gain a deeper understanding of whether astrocytomas in the thalamus possess any distinctive features, such as the capacity to induce epilepsy or other neurological disorders.
Author Response
Reviewer 1:
Dear Reviewer,
Thank you for your detailed and constructive feedback on our manuscript. We have made the following revisions based on your suggestions:
- Incidence of Thalamic Tumors in Women: We appreciate your request for additional information regarding the incidence of thalamic tumors in women. In our revision, we have included data from relevant studies.
- IDH Mutation and Thalamic Tumors: Based on your suggestion, we have expanded the discussion on the role of IDH mutations in thalamic tumors. These additional details should provide a more comprehensive understanding of the molecular landscape of thalamic tumors.
- Definition of "SVZ": We apologize for the oversight regarding the abbreviation "SVZ" (subventricular zone). This has now been clearly defined at its first mention in the manuscript to ensure clarity for the reader.
- Distinctive Features of Thalamic Astrocytomas: Following your suggestion, we have now included a section discussing the unique clinical and pathological characteristics of thalamic astrocytomas. In particular, we highlight their potential to cause neurological deficits based on the thalamus's key role in sensory and motor pathways. Additionally, although less common, some reports suggest that thalamic tumors can contribute to seizures and cognitive dysfunction, depending on tumor location and involvement of nearby structures like the corticothalamic tracts.
We hope that these revisions address your concerns and improve the clarity and depth of the manuscript.
Reviewer 2 Report
Comments and Suggestions for Authors
Dear Authors,
I have read the manuscript entitled "High-Grade Thalamic Glioma: Case Report with Literature Review" with great interest, and I am hereby sharing with you my comments.
The manuscript has several merits, with a well documented presentation of the clinical case, and an on-point review of literature. I have however a few remarks and questions I would like to bring to your attention:
1) at page 7of9 the "disclosures" section reads "Human subjects: All authors have confirmed that this study did not involve human par-ticipants or tissue", which doesn't quite read correct. In line with this, this reviewer would expect to read of an informed consent to publication signed by the patient;
2) during the description of the case, no mention is shared of the molecular profile of the lesion, despite molecular markers being mentioned in the review of literature in the "discussion"
3) in the "conclusion" a few information are shared about the clinical outcome of the case, but no information about "time" is shared with the readers (i.e. how long from symptoms onset to diagnosis; how long from diagnosis to intervention; how long was the followup after the surgery to the point of writing)
4) How did the formulated prognosis (which by the way doesn't seem to be shared with the readers) compare with outcomes?
I hope you concur that targeting the above mentioned minor issues would improve your reporting and the potential for impact of your manuscript.
Best of luck with the next steps in the publication procedure.
Author Response
Reviewer 2:
Dear Reviewer,
Thank you for your valuable feedback on our manuscript. We greatly appreciate your detailed suggestions, and we have carefully revised the manuscript to address your concerns.
- Disclosures Section: We regret the confusion regarding the statement in the "Disclosures" section. We have corrected the language to clarify that human subjects were involved, and an informed consent for publication was obtained from the patient. The section now reads: "Informed consent for the publication of this case report was obtained from the patient in accordance with ethical guidelines."
- Molecular Profile of the Lesion: You raised an important point regarding the lack of molecular profiling in the case description. We acknowledge that molecular markers are critical for understanding gliomas. We have now incorporated the molecular profile of the tumor into the case description to provide a more complete picture and align with the literature review.
- Timeline of Clinical Events: In response to your request for a more detailed timeline, we have added relevant information about the timeframes.
Reviewer 3 Report
Comments and Suggestions for Authors
The author’s contribution to the literature with this study is challenging to evaluate. The reviewer would kindly request a more extensive literature search, including all cases in Pubmed, to enrich our research and contribute to the literature.
Search terms (thalamus) AND (glioma)
Sort by most recent
https://pubmed.ncbi.nlm.nih.gov/?term=%28thalamus%29+AND+%28glioma%29&sort=date&size=200&ac=no
Please review the instructions for authors on how to re-upload the manuscript.
Author Response
Reviewer 3:
Dear Reviewer,
Thank you for your thoughtful feedback and for highlighting the need to enhance the literature review section of our manuscript. We have taken your comments into careful consideration and have made the following revisions:
- Expanded Literature Search: In response to your request, we have conducted a more extensive literature search using the PubMed database, specifically utilizing the search terms "thalamus" AND "glioma" as suggested. We have sorted the results by the most recent studies to ensure that the manuscript includes up-to-date information on the topic. The revised manuscript now incorporates a broader range of case reports, clinical studies, and reviews, which enhances the relevance and depth of the literature review.
- Enrichment of Literature Review: The newly integrated studies have contributed significantly to our discussion, particularly regarding the clinical presentation, surgical outcomes, and molecular features of thalamic gliomas. We believe that this enriched literature base will provide readers with a more comprehensive understanding of the current landscape of thalamic glioma research and its implications.
Round 2
Reviewer 3 Report
Comments and Suggestions for Authors
Satisfactory